# The “Dark Side” of Picocyanobacteria: Life as We Do Not Know It (Yet)

**DOI:** 10.3390/microorganisms10030546

**Published:** 2022-03-02

**Authors:** Cristiana Callieri, Pedro J. Cabello-Yeves, Filippo Bertoni

**Affiliations:** 1National Research Council (CNR), Water Research Institute (IRSA), 28922 Verbania, Italy; 2Evolutionary Genomics Group, Departamento de Producción Vegetal y Microbiología, Universidad Miguel Hernández, 03550 San Juan de Alicante, Spain; pedrito91vlc@gmail.com; 3Museum für Naturkunde, 10115 Berlin, Germany; bertonifilo@gmail.com

**Keywords:** Black Sea, mesopelagic zone, picocyanobacteria, *Synechococcus*

## Abstract

Picocyanobacteria of the genus *Synechococcus* (together with *Cyanobium* and *Prochlorococcus*) have captured the attention of microbial ecologists since their description in the 1970s. These pico-sized microorganisms are ubiquitous in aquatic environments and are known to be some of the most ancient and adaptable primary producers. Yet, it was only recently, and thanks to developments in molecular biology and in the understanding of gene sequences and genomes, that we could shed light on the depth of the connection between their evolution and the history of life on the planet. Here, we briefly review the current understanding of these small prokaryotic cells, from their physiological features to their role and dynamics in different aquatic environments, focussing particularly on the still poorly understood ability of picocyanobacteria to adapt to dark conditions. While the recent discovery of *Synechococcus* strains able to survive in the deep Black Sea highlights how adaptable picocyanobacteria can be, it also raises more questions—showing how much we still do not know about microbial life. Using available information from brackish Black Sea strains able to perform and survive in dark (anoxic) conditions, we illustrate how adaptation to narrow ecological niches interacts with gene evolution and metabolic capacity.

## 1. Introduction: Understanding Picocyanobacteria

The term picocyanobacteria refers to prokaryotic autotrophic microorganisms in the size scale of picoplankton (0.2–2 µm), small even for microbial life [1]. Their diminutive size and relatively large surface area make picocyanobacteria particularly efficient and resilient and, in combination with their versatile molecular machinery, can help account for their global distribution and success [2]: these microorganisms can be found virtually in every aquatic environment, from marine, to brackish, to freshwater systems, at both cold and warm temperatures, different light conditions, and in a variety of nutrient concentrations [3,4]. Since their early characterization [5], picocyanobacteria revealed previously unexpected dynamics, especially in the ocean, where the ecological importance of their contribution to carbon cycling and primary production quickly became apparent. Nevertheless, it was only from the 2000s, as molecular techniques and novel computational approaches became more accessible, that scientists could zoom into the metabolic and genetic machinery that has made cyanobacteria so successful, shining new light over their long evolutionary history. Over the last decades, we learned how, since their emergence—likely at the end of the Pre-Cambrian [6]—picocyanobacteria have significantly contributed to Earth’s biogeochemical cycles [7] and even played a central role in the evolution of eukaryotic photosynthesis [8]. Thanks to their versatility, adaptability, and resilience, picocyanobacteria are widely distributed and abundant and deeply impact our planet’s biosphere. For all these reasons, the continued study of these small microorganisms promises to help us better characterize ecosystem function from the pico-sized scale all the way to the planetary scale.

To contribute to this effort, in this article we review our current understanding of picocyanobacteria. We do so by focusing on the genus *Synechococcus*, one of the most widespread and cosmopolitan picocyanobacterial genera [9,10,11]. Starting from the complex phylogenetic relations behind this genus and other picocyanobacterial lineages, we review current knowledge on the unique physiological features that characterize this genus, as well as its role and distribution globally in the wide range of environments it inhabits. As one of the best-studied genera of picocyanobacteria, *Synechococcus* is a good representative of these pico-sized microorganisms. Nevertheless, as we show in the third part of this article, even such a well-characterized genus still holds many surprises. This becomes clear in the case study following the unexpected recovery of *Synechococcus* in the deep dark anoxic layers of the Black Sea: even in the extreme conditions of this environment, picocyanobacteria manage to find ways to survive [12]. With the help of genomics and other molecular approaches, we are beginning to shed light on the metabolic and genetic machinery that makes this impressive adaptability possible, as the case of picocyanobacteria in the mesopelagic Black Sea shows. In this sense, this article offers a review of what we know about picocyanobacteria, but also an important reminder of how much we still do not know.

## 2. The Genus Synechococcus

### 2.1. Phylogenetics: A Cryptic Genus?

Picocyanobacteria of the genus *Synechococcus* include a diversity of cosmopolitan photoautotrophic prokaryotes which can be found as single-cells or as microcolonies in a wide range of aquatic environments [3,4]. Its diversity, which allowed *Synechococcus* to colonise such different environments, in combination with the genus’ potential to fix significant amounts of CO_2_, attracted the attention of microbial ecologists [13,14]. For this reason, together with *Cyanobium* (in freshwaters and brackish environments) and *Prochlorococcus* (in oceans), *Synechococcus* is among the most well-characterized picocyanobacterial lineages, the smallest organism within the taxa of cyanobacteria [15], making it a useful representative genus for this review. *Synechococcus* has been considered as a coherent genus because of its low morphological variability [16]. However, the genus is known to be polyphyletic [9], and various molecular techniques have highlighted significant differences between strains [17,18,19]. Since the early 2000s, the majority of *Synechococcus* isolates have been taxonomically placed into a cyanobacterial lineage termed cluster 5 [10], sometimes called the ProSyn clade after its main representatives, *Prochlorococcus* and *Synechococcus*. Although, as the analysis of full genomes has demonstrated that studies based solely on the 16S rRNA gene have relatively low resolution to discriminate between picocyanobacteria strains, the details of the phylogeny of this genus could start to emerge only over the last decade [20,21,22,23]. Importantly, the recent isolation and sequencing of new marine, brackish, and freshwater strains have increased the resolution of picocyanobacteria’ phylogenetics and have revealed unexpected relationships between strains from different environments [22,24,25,26]. These studies have allowed us to trace a more accurate phylogenomic tree that places all culture-derived picocyanobacteria from the genera *Synechococcus* and *Cyanobium* inside three sub-clusters (SC), SC5.1, SC5.2, and SC5.3 (Figure 1). Within this subdivision, SC5.1 is traditionally considered as mainly marine, but it also includes some brackish/halotolerant strains, such as some Red Sea and Black Sea isolates (BS55D/BS56D). Most other brackish/halotolerant strains are placed inside SC5.2 (e.g., WH5701 or Black Sea BSF8S/BSA11S). They share this subcluster with some freshwater strains, with *Cyanobium usitatum* [22], *Vulcanococcus limneticus* [27], and *Cyanobium gracile* and other *Cyanobium* spp. as main groups. However, the majority of freshwater strains affiliate inside SC5.3, with *Synechococcus lacustris* isolates as representative species [21,22]. Perhaps unsurprisingly, SC5.2 is the most insightful subcluster in terms of genomic diversity, presenting strains that span a wide range of salinity adaptation, from the truly freshwater environment to estuarine/brackish and, ultimately, marine environments. A recent proposed systematization offers to rename the members of the subclusters as Ca. Marinosynechococcus (SC5.1), *Cyanobium* (SC5.2), and Ca. Juxtasynechococcus (SC5.3) [23], however, the taxonomic resolution of these lineages is still poor, even if new isolation and sequencing campaigns of freshwater and brackish isolates that are underway promise to complete the general phylogenetic trend that is currently available and to allow further comparisons along the salinity divide [28]. For this reason, we will use “*Synechococcus*” throughout the text to refer both to cluster 5 picocyanobacteria and other taxa such as *Synechococcus elongatus* or *Synechococcus*-like isolates.

### 2.2. Physiological Features

Picocyanobacteria’s tiny cells (e.g., *Synechococcus*, around 2 µm) resemble typical Gram-negative bacteria in structure, but present some differences that account for picocyanobacteria’s unique features. The outer and inner membranes form a cushion of periplasm with a thick peptidoglycan layer. They constitute a mechanical and permeability barrier for most large molecules, resulting in high protection from harmful agents [29]. This layer of peptidoglycan is thicker than that of most Gram-negative bacteria and is complexed with polysaccharides, resembling more the peptidoglycan of Gram-positive bacteria. Furthermore, the outer membrane is composed of lipopolysaccharides with bound phosphate and contains unusual fatty acids, carotenoids, and porins that may be the results of specific adaptations during *Synechococcus’* long evolution. Externally, some *Synechococcus* spp. can form a surface S-layer of glycoproteins that covers a multi-layered envelope (Figure 2). The S-layer is a crystalline proteinaceous structure often covered by a mucilaginous sheath and formed by hexagonal monomers that constitute a hexagonal lattice on the cell envelope; these structures are generally involved in cell adhesion and protection and act as molecular sieves [30,31]. In addition, this layer has been associated with swimming capability in marine *Synechococcus* spp., as it generates thrust by means of rotation [32,33]. The creation of thrust is due to the presence of a cell-surface-associated polypeptide present in the S-layer of marine *Synechococcus* sp. strain WH8102 and encoded by the gene *swm*A [33,34]. In addition to *swm*A, another large cell-surface protein, *swm*B, is involved in motility, but its role and connection with the S-layer still eludes researchers [35,36]. Other structures tied to the cell surface are extracellular appendages such as the pili [37] and the spinae [38]. The type IV pili are appendages that can be extended and retracted, not only for motility or to attach to a surface and form a biofilm, but also to float in planktonic lifestyles and to avoid predation [39]. The spinae are rigid tubes from 100 nm to 1 µm long that can induce microcolony formation in freshwater *Cyanobium* sp. [38]. Furthermore, the S-layers of *Synechococcus* can have another important function as they template calcite nucleation by binding Ca^2+^ to the outer surface of the S-layer that increases the pH in the region surrounding the cell [40,41], causing so-called whiting events [42]. During these events, observed in many lakes and coastal areas, CaCO_3_ in the form of calcite precipitates on the S-layer membranes, which are continuously shed and replaced, allowing the precipitation cycle to be continuously repeated [43]. Interestingly, *Synechococcus’* S-layers can also promote the immobilization of Sr^2+^ when it is the major divalent cation present, biogenically forming celestite and strontianite, previously thought to be purely evaporitic minerals [44]. The presence of *Synechococcus* also promotes Mg carbonate precipitation as hydromagnesite by increasing fluid pH and, potentially, by providing nucleation sites at the cell surface for mineral growth [45].

In addition to these structures, *Synechococcus* holds the largest pigment diversity within autotrophs, which allows these organisms to occupy a large range of light niches of different aquatic habitats [47,48,49]. The structure that accommodates many of these pigments is the phycobilisome (PBS). PBSs are the major antenna protein complex of cyanobacteria, used to harvest light in the 500–650 nm (green to orange) range, while chlorophyll *a* (Chl *a*) absorbs mainly blue (440 nm) and red light (678 nm). The PBSs are located on the thylakoid membranes, are composed of phycobiliproteins, and consist of a core made of allophycocyanin, from which stacked rods extend outwards, comprising phycocyanin at the base, followed by chromophores of phycoerythrin (PE) and/or phycocyanin (PC) [48,50] (Figure 3). PE absorbs light at a peak of ~570 nm, corresponding to the green part of the spectrum, and PC absorbs at a peak of ~630 nm, in the red part of the spectrum [48,51], although variations can exist within these two main types. Correspondingly, PE-rich picocyanobacteria are often found in clear, oligotrophic waters where green and blue-green light is available, while PC-rich phenotypes are found in more turbid, eutrophic waters dominated by red underwater light [52,53,54] (Figure 4). The presence of light-absorbing pigments at different wavelengths extends the occupancy of ecological niches with different light characteristics in the water column (e.g.,) [52,54,55,56]. Thus, the PBS, with its antenna pigments capable of using different wavelengths, is an advantage for *Synechococcus*, providing a functional basis for their adaptation to different underwater radiation conditions. However, *Synechococcus* decreases its photosynthetic efficiency in predominantly blue light environments, as the phycobilin pigments mainly do not absorb photons in the violet/blue part of the spectrum (≤450 nm), with the exception of phycourobilin [57]. Specific studies have shown the PBS’s poor absorption of blue light [48] to be one of the major selective advantages of *Prochlorococcus* over *Synechococcus* in colonizing deep oceanic waters, as *Prochlorococcus* lacks the PBS and hosts instead divinyl-Chlorophyll *a* and *b*, thus evolving light-harvesting antennae that effectively absorb blue light [58]. Consequently, *Synechococcus* usually thrives in the near-surface layers of the oceans, in coastal waters and lakes, where green and blue-green light is available [54,59]. In Figure 3, the simplified scheme of the photosynthetic machinery responsible for the process of converting solar energy into biomass shows three protein complexes immersed in the thylakoid membrane: photosystem I (PSI), photosystem II (PSII), and cytochrome b6f (Cyt b6f). Most PBSs are typically associated with PSII, while Chl *a* and carotenoids are mainly bound to PSI. It has long been assumed that the light energy absorbed by the PBS is transferred via allophycocyanin to the chlorophyll *a* present in PSII [60] and successively passed to PSI through what is called spill-over [61]. However, it is now established that, depending on the prevalent wavelengths, the PBS can move between PSII and PSI in a process called state transition [62]. The picture that emerges when considering all these different features of picocyanobacteria’s cells is that of microorganisms that are very well protected from environmental changes, have unexpected motility, aggregation and biogenic capacities, and a broad range of light-harvesting structures that can partly explain the great resilience of this genus.

### 2.3. Role and Distribution

In the effort to understand the relationships between *Synechococcus* abundance and chemical, physical, or biological factors that shape its dynamics in aquatic environments, two different trends can be observed. On the one hand, oceanographers have concentrated on the spatial dimension by relying on transects across the vast expanse of the Earth’s oceans; on the other hand, limnologists, whose environments of study tend to be easier to repeatedly access and sample, have focused on the temporal dimension of *Synechococcus* distribution, observing the seasonal dynamics of these organisms. Despite this underlying difference, both groups of researchers focused on two factors that are fundamental to the life and growth of most autotrophic organisms: temperature and nutrients.

#### 2.3.1. Temperature

In the sea, *Synechococcus* is present in waters from low (2–3 °C) to high (>30 °C) temperatures. However, a positive relationship between mean annual abundance and temperature has been found below 14°C, as above that temperature nitrate concentrations are usually very low and may therefore replace temperature as the most significant driver of picocyanobacteria abundance [4,63]. In Lake Maggiore and other temperate lakes, the peak abundance of picocyanobacteria is found at the isotherms with temperatures between 18 °C and 20 °C [64,65], where the thermal conditions are important not only for the ambient water temperature per se, but also for the maintenance of a density-gradient resisting settlement [66] and for the stability of lake vertical structure, which impacts picocyanobacteria abundance [67]. Nevertheless, the presence of picocyanobacteria at lower temperatures, observed in many marine environments, was not uncommon even in freshwater systems. For example, in Lake Maggiore, *Synechococcus* was found deep in the water column at temperatures lower than 10 °C, as well as in North Patagonian ultraoligotrophic lakes [68] and in the Russian Lake Baikal [69]. It must, ultimately, be considered that temperature is linked to seasonality, and it is therefore difficult to separate this variable from the many biotic and abiotic factors linked to it that may or may not have a direct influence on *Synechococcus’* abundance.

In marine environments, the distribution of *Synechococcus* in relation to temperature was obviously linked to the presence of these microorganisms at different latitudes. For example, in both the South and North Atlantic Oceans, where temperatures can reach below 0 °C, a decrease in cell numbers was observed [70,71]. The presence of *Synechococcus* in the Arctic Ocean indicates that these cells can survive in the −1.8 °C/4.1 °C temperature range, but a possible transport of *Synechococcus* from areas of the Pacific Ocean or its release through rivers could explain its recovery in these extreme environmental conditions [72,73]. In a rare study covering an annual cycle within the Atlantic inflow to the Arctic Ocean [74], cold-adapted *Synechococcus* were found as far as 82.5°N, with numbers in the range of 600–21,300 cell ml^−1^ at 79°N, in November and August, respectively. In the cold bathypelagic Black Sea (around 1000 m deep), *Synechococcus* has been recently detected [12,75], and its vertical distribution showed an increase in picocyanobacteria numbers both in West and East deep stations [24] (Figure 5), as we will further explore later.

Furthermore, there are marine environments where, even at low temperatures, there can be relevant abundances of *Synechococcus*, such as in the Baltic Sea, where this genus can reach 10^5^–10^6^ cells ml^−1^ [76]. A recent estimate of the global distribution of *Synechococcus* in the oceans, based on more than 37,000 observations, predicts 7.0 ± 0.3 × 10^26^ cells as the annual mean [14]. Using niche models, the same authors predicted that *Synechococcus* was mostly abundant in the Indian and western Pacific Oceans (maximum = 3.4 × 10^4^ and 4.0 × 10^4^ cells mL^−1^, respectively), peaked at mid-latitudes, and had low abundance rates in the Arctic and Southern Oceans [14]. Unfortunately, there are no corresponding estimates at the global scale for *Synechococcus’* annual mean abundances in freshwater systems. Nevertheless, reviewing literature on *Synechococcus* abundance estimates in freshwaters, and considering lakes at different trophic states, the number of *Synechococcus* can be observed to range from 10^3^ to 10^7^ cells ml^−1^ [66]. A detailed review on autotrophic picoplankton, which includes picocyanobacteria, reports the range of abundances from shallow turbid lakes to clear ultraoligotrophic lakes in the southern hemisphere (Argentina), with a maximum of 2 × 10^7^ cells ml^−1^ [77]. The presence of microcolonies can largely increase the total *Synechococcus* numbers, as found in many lakes in Argentina [77], in Hungary [78], and in New Zealand [79] characterized by the dominance of PC cells. High numbers have been also reported in Mexico in some volcanic alkaline lakes [80] and in Spanish karstic lakes, where *Synechococcus* can reach several millions of cells per ml [67,81].

#### 2.3.2. Nutrients

The other factor studied as an important driver for *Synechococcus* abundance in natural systems is nutrient concentration. A first rough generalisation marks a difference between marine and freshwater systems: nitrogen and iron are, in general, the main limiting nutrients for growth and production in the oceans [82], whereas in lakes phosphorus has been found to be the limiting nutrient [83]. Yet, there are several exceptions to this rule of thumb, as in the case of Argentinian lakes where Diaz and co-authors [84] demonstrated that nitrogen deficiency, more than phosphorus, can limit the production in ultraoligotrophic Patagonian lakes. Similarly, in marine systems in conditions of P-limitation, the capability to opportunistically use organic P can vary among different eco/genotypes of picocyanobacteria [85]. These exceptions remind us of the importance of looking more closely at paradigmatic assumptions about the differential role of nutrients in different environments, especially in relation to the use of nutrients by microorganisms.

Several laboratory studies have been performed to test the growth ability of *Synechococcus* in conditions of different limiting nutrients (extensively reviewed in [3] for freshwater strains). There is evidence that ammonium is the preferred form of nitrogen for *Synechococcus* in culture [86,87], but when ammonium is exhausted, *Synechococcus* can utilize nitrate, thanks to a regulatory mechanism that can induce expression of nitrate reductases [88]. Furthermore, under severe inorganic N-limitation, urea present in the dissolved organic matter can become the main N source for cyanobacteria [89] and for *Synechococcus* [90]. As for P, in addition to using orthophosphate like all other autotrophic cells, *Synechococcus* can also hydrolyse organic phosphorus, thanks to extracellular phosphatase activity [91,92].

In both oceans and lakes, the number of *Synechococcus* increases with increasing trophy, as do many other microalgae and cyanobacteria. However, the relative importance of *Synechococcus* numbers and biomass to total phytoplankton decreases with trophy. This model, introduced by Stockner [93], has been much discussed over the last twenty years and has found both supporters and detractors [52,78,94,95,96,97]. What is certain is that in oligotrophic systems, where competition for nutrients is high, organisms that can grow using different nutritional sources have the upper hand. This is also the case for the small *Synechococcus* cell, which has less need for nutrients than other algae, both because of the simple structure of the prokaryotic cell compared to the eukaryotic one and because of its ability to activate symbiotic cooperation with other organisms [98].

#### 2.3.3. Primary Production

Despite the abundance of data on the contribution of the picoplanktonic fraction to total primary production (e.g.,) [64,99,100,101,102], these overestimate the contribution of picocyanobacteria as they also include the production of picoeukaryotes, which may be significant in lakes or coastal waters. It is important to stress that to quantify the contribution of *Synechococcus* to total CO_2_ fixation, it is necessary to use radiochemical methods coupled with flow cytometric sorting to attribute NaH^14^CO_3_ incorporation to specific groups; to date, there are only few studies that have used this elaborate method [13,103]. According to these studies, in subtropical North Atlantic waters, *Synechococcus* has been estimated to contribute 21% to total CO_2_ fixation [13]. In lakes, CO_2_ incorporation by the picoplanktonic fraction, mainly composed by picocyanobacteria, especially in oligotrophic conditions, ranges from very low in winter to a maximum of 13 or 7.5 mgCm^−3^ h^−1^ in the summer (in Lake Maggiore and Lake Constance, respectively [104]). The percentage of primary production of picocyanobacteria on the total production by phytoplankton ranged from 10 to 21% in the meso-oligotrophic Lake Maggiore, while it reached up to 80% in the ultra-oligotrophic Lake Baikal [100].

Since the early 1980s, the presence of autofluorescent pigments has made it possible to divide *Synechococcus* into cells with phycoerythrin (PE) [105] and cells with phycocyanin (PC), even by simple epifluorescence microscope observations [106]. This rough subdivision allowed observation of an important relationship between *Synechococcus* types and underwater light quality. It was found that PEs were more abundant in oligotrophic and transparent environments with prevailing green and blue light, while PCs were predominant in eutrophic systems with red light. However, when studying the pigments in more detail, as well as their associated genes, this subdivision appears reductive, as exemplified in recent papers demonstrating the presence of three types of pigment clusters with the new Type IIB [25,107].

#### 2.3.4. Community Composition

Perhaps the most interesting advance in the study of *Synechococcus* dynamics is the possibility to distinguish between different genotypes and to trace community composition based on different genes or on the entire genome. Different molecular techniques with increasing descriptive power and reliability are now in common use in field studies. For example: clone library sequencing (e.g., [108]), terminal restriction fragment length polymorphism analysis of the 16S–23S rRNA internal transcribed spacer region (e.g., [109]), specific gene markers analysis by qPCR (*rpoC1*, *petB*, *cpcBA, cpeBA* ecc.) (e.g., [27,110,111,112]), as well as metagenomic (recruitment) analysis [22,23,25,28]. The studies carried out using these techniques made it possible to obtain a dynamic distribution not only of the number of picocyanobacteria, but also of strains or clades. This provides information on certain strains’ preferences for different ecological niches. For example, the studies on adaptation and acclimation capacities to light suggest that *Synechococcus* can grow over large ranges of irradiance [113,114] and that a high functional microdiversity exists also in relation to their capacities for acclimation to temperature [115] and salinity [116].

The spatial biogeography of *Synechococcus* in the oceans has shown the domination of clade I and IV in temperate, coastal waters [117] and of clade II in tropical and offshore oligotrophic systems [118]. Nevertheless, their dominant role in the different systems varied during the season, as demonstrated by the long-term monitoring of a coastal station in the Pacific Ocean [112]. Picocyanobacteria from temperate lakes have shown seasonal cycles, with a spring and a summer peak [65,119] or only one peak in the summer [120], with picocyanobacterial clades changing over an annual cycle in two man-made lakes in UK [121]. The temporal change in picocyanobacteria community composition was found in Lake Constance [122], in Lake Superior [123], in Lake Balaton [78], and in Lake Maggiore [124]. In this last deep lake, different ecotypes were also shown to coexist along vertical gradients, as they rapidly acclimated and performed differently in different microhabitats.

## 3. In the Dark: *Synechococcus* Surprising Adaptability

### 3.1. A Case Study: The Mesopelagic Zone of the Black Sea

Traditionally, the study of picocyanobacteria has been limited to the euphotic zone, since these microorganisms, as photoautotrophs, depend on light for energy. For this reason, most studies of *Synechococcus* concentrated on determining this genus’ relationships with physicochemical conditions associated with trophic state at high numbers or at peak production. This approach has allowed researchers to understand many aspects of *Synechococcus* ecology and physiology. However, in the 1980s, *Synechococcus* had already been recovered from low-temperature and light-deprived conditions. The occurrence of *Synechococcus* in Antarctic waters was the first evidence of their ability to survive at low temperature [125], a finding later compounded by their recovery in the Arctic, as well. In this hostile environment, *Synechococcus* numbers were very low (40–80 cells ml^−1^), but remained constant in summer and winter, reflecting this organism’s ability to maintain growth even in the winter [73]. These unexpected findings were not limited to high latitudes: low numbers of *Synechococcus* cells (2.4–190 cells ml^−1^) have also been found in subtropical NW Pacific in meso- and bathypelagic waters, transported from the epipelagic zone down the water column [126]. These findings revealed how photoheterotrophy, which is the ability to utilize organic substrates and to harvest light energy [127,128], was therefore not just confined to proteorhodopsin-containing bacteria and aerobic anoxygenic phototrophic bacteria but was also possible for picocyanobacteria in Arctic waters [129]. Subsequent studies have demonstrated the ability of *Synechococcus* not only to utilise amino acids and glucose but especially to combine photoautotrophy with heterotrophy, typical of mixotrophic organisms [130,131]. The presence of genes encoding amino acid, oligopeptide, and sugar transporters [50] suggested their ability to acquire alternative forms of carbon and perform a combination of different nutritional pathways. The characterization of picocyanobacteria’s extensive metabolic capabilities made it possible to begin understanding why these organisms were found in deep waters where photoautotrophic life is precluded [2,73,130].

Indeed, *Synechococcus* has been found in various aquatic environments at depths greater than 200 m, in light-deprived conditions marked by low temperatures and high latitudes. The recent isolation of two *Synechococcus* strains from a depth of 750 m in the anoxic zone of the Black Sea corroborated these observations [12]. Furthermore, the mesopelagic Black Sea shows a combination of low temperatures and aphotic conditions with a complete absence of oxygen and abundance of hydrogen sulphide [132], as well as an intermediate salinity (18–22 gL^−1^), which make it a particularly harsh environment—a present day analogue for the Proterozoic ocean. This unexpected finding was prompted by the chlorophyll *a* concentration increase at between 100 and 1000 m depth (around 0.2–0.3 µg L^−1^), measured by several Bio-Argo profiling floats. This signal seemed consistent with the so-called “deep red fluorescence”, sometimes associated with CDOM (chromophoric dissolved organic matter) [133]. However, the cytometric count (up to 10^3^ cells ml^−1^) (Figure 6), the observations at the microscope (Figure 7), and the metagenome analyses [75] unequivocally demonstrated the presence of picocyanobacteria [12], supporting previous speculations that the unusual chlorophyll signal might indicate viable populations of PE-rich photoautotrophic cyanobacteria [134].

The molecular analysis of the picocyanobacterial strains isolated from the Black Sea at 750 m (BSD: BS55D and BS56D) and 5 m (BSS: BSA11S and BSF8S) showed that the isolates from the mesopelagic zone differed phylogenetically from those from the epipelagic zone [24]. Both strains possess the machinery for the protochlorophyllide reduction to chlorophyllide *a*, with one light-independent polypeptide oxidoreductase variant (DPOR). The activity of the light-independent variant for dark Chl *a* biosynthesis might explain the relationship between picocyanobacteria and the Chl *a* increase in deep dark waters. Through dedicated experiments, the ability of *Synechococcus* BS56D to synthesize Chl *a* in the dark was eventually demonstrated [12]. The experiments, together with the genomic study of BS56D, consolidated the hypothesis that the unexpected Chl *a* concentration emerging from in situ profiles might be explained, at least partially, by the presence of *Synechococcus* spp. in the mesopelagic and bathypelagic realm of the Black Sea, up to now considered unsuitable for the survival of photoautotrophic cells.

### 3.2. Unexpected Metabolisms

This left us with the question of how these picocyanobacteria could live at 750 m in the absence of oxygen and light. Studies of the genome of the isolated strains and unpublished experiments (Cabello-Yeves et al. in preparation) have suggested that they can carry out mixed acid fermentation in addition to the already known ability of mixotrophy. The presence of the photosynthetic apparatus and pigments in the PBS, with their unaltered fluorescence, were interpreted as a reserve ready for eventual use in case of resuspension or transport in more superficial layers, even with very little light. Importantly, the isolated strains were able to resume photosynthetic and growth functions as soon as they were placed in the right conditions. The qPCR assays on DNA from Black Sea samples from two vertical profiles in the western and eastern deep part of the Black Sea and in coastal superficial sites, using strain-specific primers targeting *rpoC1* genes, showed a different distribution of the BSD and BSS. The BSD strains originate from the upper epipelagic waters but can cope with the harsh conditions of the deep layer, displaying higher resilience and adaptability, while coastal BSS strains were specifically attuned only to coastal epipelagic waters [24].

The potential adaptation of these small phototrophs to adverse conditions such as those observed in the meso-bathypelagic dark anoxic Black Sea was also assessed thanks to the presence of several genes, indicating that these molecular tools could be used for their survival in dark anoxic systems. At the genomic level, the BSD and BSS strains showed several differences in terms of phylogenetic placement, genome size, and genomic content. For instance, deep ecotypes (BSD) affiliated within the clade VIII/IX inside SC5.1 (Figure 1) had a smaller genome (ca 2.3 Mb) than the surface strains (BSS) (ca. 3 Mb), which affiliated inside SC5.2 proximal to other euryhaline/brackish strains. With regard to gene content, a noteworthy feature observed in all these Black Sea isolates was their ability to conduct mixed-acid fermentations. Deep strains harboured the catabolic acetolactate decarboxylase specific for acetoin fermentation and D-lactate dehydrogenase for lactate fermentation [12,24]. BSF8S/BSA11 harboured three different lactate dehydrogenases for lactate fermentation, together with hydrogen production via bidirectional hydrogenase (hox genes). Furthermore, they possess a potential formate hydrogenlyase 3 (HyfB- COG0651), hydrogen uptake hydrogenase (hyp genes), or a V-type ATP synthase next to one of those lactate dehydrogenases, all of which are genomic tools which would help these strains’ performance in anoxic environments [24].

### 3.3. Ecological Considerations

The case study of *Synechococcus* in the Black Sea’s deep waters demonstrates that, in anoxic dark conditions, the BS56D strain accumulates chlorophyll *a* thanks to the presence of different enzymes from the anaerobic pathways of the chlorophyll *a* biosynthesis and has all the machinery to perform fermentation. On the one hand, this discovery provides new insight to support early speculations that associated the origin of the “deep red fluorescence” signal to viable picocyanobacteria populations in deep dark anoxic waters. On the other hand, the case works as a reminder that what we think we know of life might sometimes get in the way of understanding microbial dynamics. While, certainly, *Synechococcus* thrives in the euphotic zone where its peak abundances are recorded, this should not hinder us from exploring and understanding its presence in deep dark waters. Even photosynthetic organisms such as the picocyanobacteria we described can be found in dark anoxic water, using alternative metabolisms to photosynthesis that allow them to survive, albeit with low growth rates, in hostile and lightless environments. The viable stock of *Synechococcus* found in anoxic mesopelagic waters was in fact able to retain its photosynthetic ability in case of re-establishment in the upper layer.

This is especially important in the case of *Synechococcus*, given the evolutionary, molecular, and ecological role of these picocyanobacteria. While the adaptations found in the Black Sea strains might not be directly involved in their current success and contribution to planetary biogeochemistry, they could still reveal unexpected and surprising pathways that might have had an important role over the long evolutionary history of this genus. This is particularly true in the Black Sea, which has already been indicated as an analogue for the Proterozoic ocean, and whose current dynamics—even in its most obscure aspects—might provide useful insights into the oxygenation processes that have shaped the planet, processes that are all the more important to understand if we are to face the current climatic changes. In this sense, the case study we offered here also reminds us that—against the tendency, until recently still common among environmental microbiologists, to focus on highly abundant, widespread, or significant groups—even rarer species or uncommon ecotypes might have important implications and might be crucial to understanding Earth’s history and its current dynamics. The low numbers of picocyanobacteria found in the mesopelagic zone of the Black Sea do not make this finding less significant. In fact, if we consider these ecotypes and their unusual adaptations as rare, we would do better by taking them seriously and studying them even closer; their molecular makeup might hold clues to important biogeochemical processes that are still poorly understood. It is precisely through the study of rare species that we can try to unravel the skein of the planet’s evolution by characterizing the dynamics and evolution of the microorganisms that helped shape it [135].

## 4. Conclusions: Learning from Life as We Do Not Know It

In this article, we reviewed our current understanding of picocyanobacteria by focusing on the genus *Synechococcus*. Thanks to their diversified and complex metabolic machinery, these tiny prokaryotic cells have come to colonize virtually all aquatic environments and significantly impact global dynamics as well as evolutionary history. However, while there is much that is known about picocyanobacteria of this genus, our review also highlights that there is still much yet to be understood. The impressive features of these microorganisms keep challenging our disciplinary approaches and provide the impetus to rethink long-held assumptions. Even a genus as relatively well-characterized as *Synechococcus*, as we showed, still holds surprises—such as its ability to adapt to the dark anoxic deep waters of the Black Sea. With the help of novel molecular and computational technologies, we can now begin to shed light on these often-unexpected features. This, we suggest, promises to help us better resolve the dynamic interactions between the pico-scale and the planetary scale, articulating the relationship between adaptations to different ecological niches and the evolution of genes and metabolic pathways. Perhaps the key to *Synechococcus’* success, its adaptability, can also provide us with clues for better understanding not only life as we know it, but also life as we do not know it (yet).

## Figures and Tables

**Figure 1 microorganisms-10-00546-f001:**
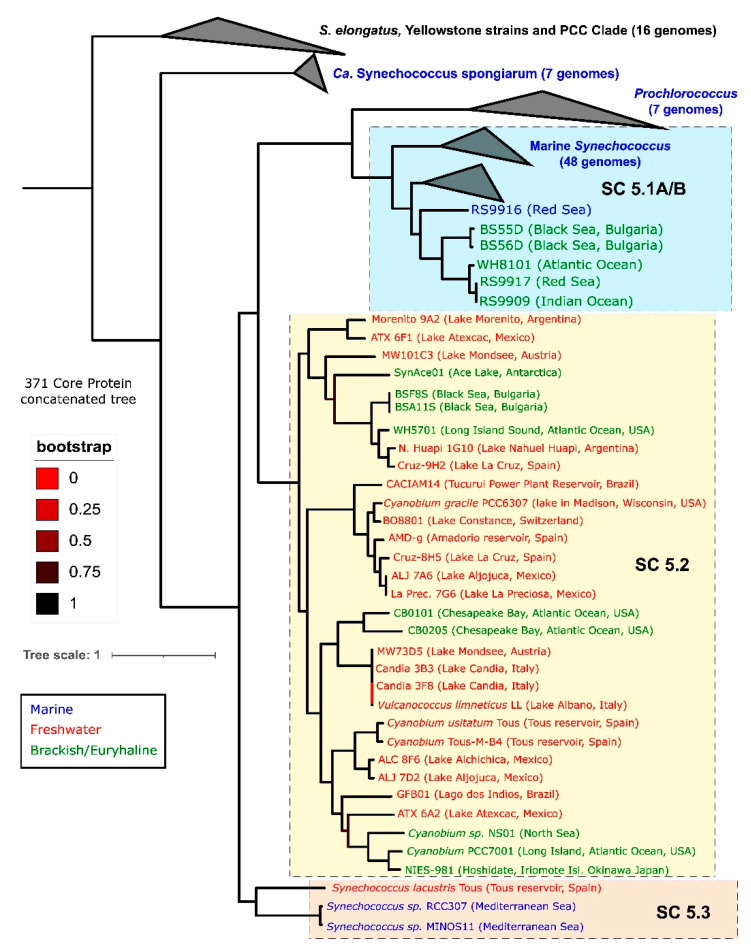
Protein-concatenated phylogenomic tree constructed with 371 core proteins with PhyloPhlAn3 tool. The tree includes all culture-derived picocyanobacteria from *Synechococcus* and *Cyanobium* genera inside sub-clusters 5.1, 5.2, and 5.3, a few *Prochlorococcus* representatives and Ca. *Synechococcus spongiarum*. The origin of each isolate is colour coded. The tree was rooted at the *S. elongatus* and PCC clade. Bootstrap values higher than 0.95 are marked as black squares on nodes. Modified from [26].

**Figure 2 microorganisms-10-00546-f002:**
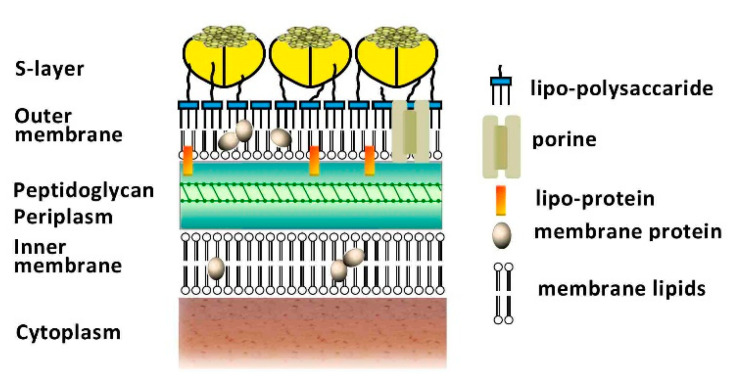
Typical structure of the envelope of a cyanobacterium (inspired from) [31,40,46].

**Figure 3 microorganisms-10-00546-f003:**
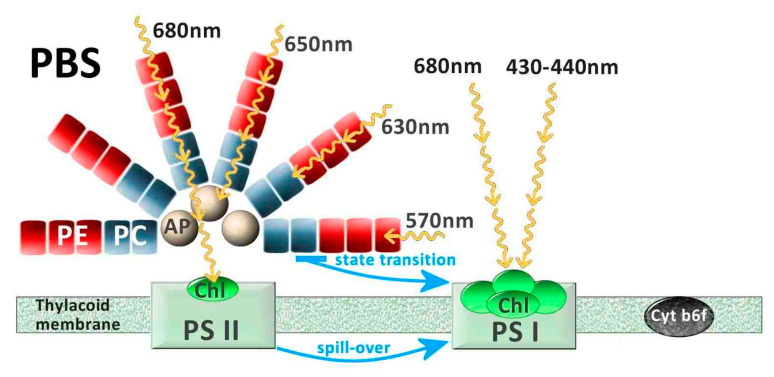
Structure of the photosynthetic machinery in picocyanobacteria. The yellow arrows indicate the different wavelengths captured by phycobilisome (PBS) pigments and by chlorophyll *a* (Chl). The energy transfer can be carried out through spill-over from PSII (photosystem II) to PSI (photosystem I) or through state transition due to direct acquisition from PBS by PSI. AP (allophycocyanin), PC (phycocyanin), PE (phycoerythrin), Cyt b6f (cytochrome b6f).

**Figure 4 microorganisms-10-00546-f004:**
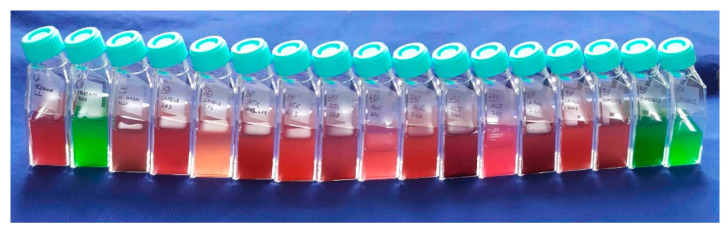
Cultures of 17 freshwater strains of *Synechococcus* with different pigments that show the diversity in pigment composition with different pink, yellowish, and green colours (from [26], modified).

**Figure 5 microorganisms-10-00546-f005:**
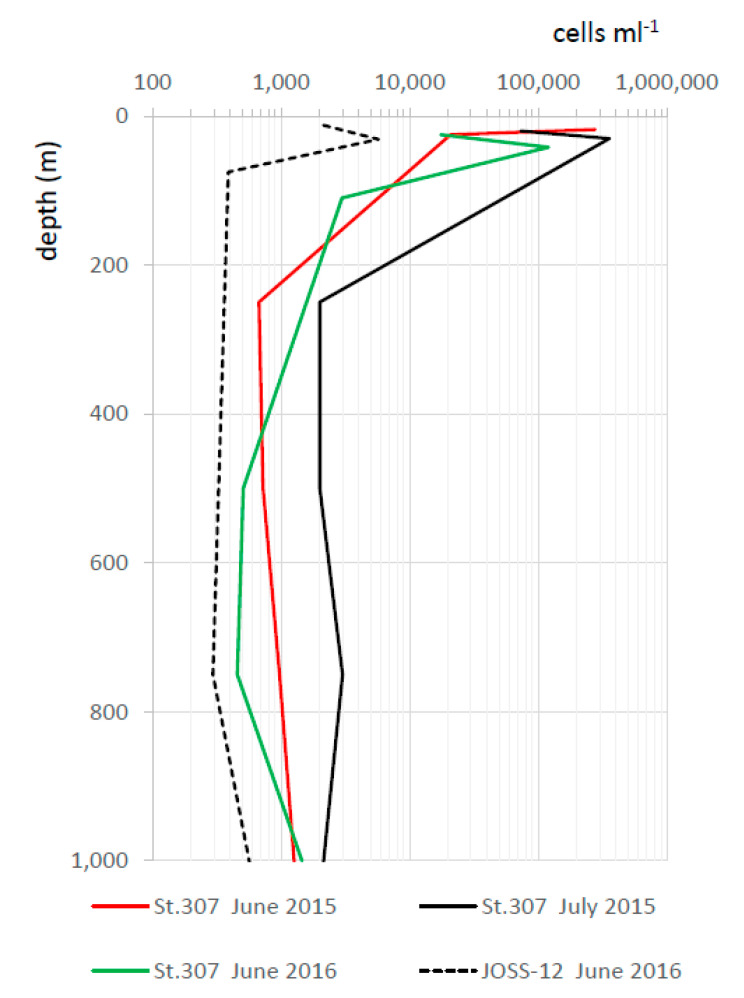
Vertical distribution of picocyanobacteria number in the West (St. 307) and East (JOSS-12) station of Black Sea (from [12,24], modified).

**Figure 6 microorganisms-10-00546-f006:**
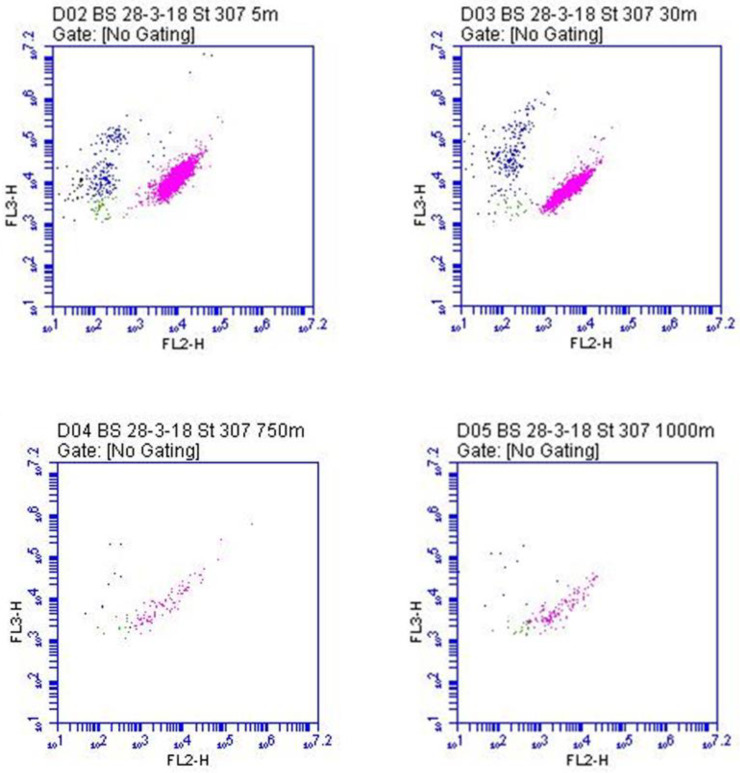
Cytograms of samples from different depths of the Black Sea (for methods, refer to [12]), (Cytometer Accuri C6, Becton Dickinson, Oxford, UK). In the density plots the pink dots are the picocyanobacteria. Orange fluorescence: FL2 channel = 585/40 nm and red fluorescence: FL3 channel >670 nm.

**Figure 7 microorganisms-10-00546-f007:**
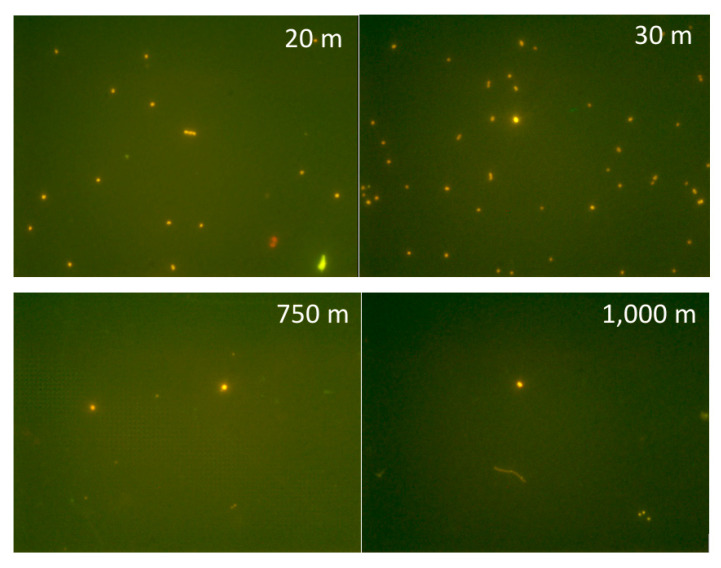
Picocyanobacteria from different depths (20, 30, 750, 1000 m), Black Sea, station 307, (epifluorescence microscopy, Zeiss Axioplan, 1250×, blue filters).

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
