# Peer review of "The “Dark Side” of Picocyanobacteria: Life as We Do Not Know It (Yet)"

_microorganisms, 2022, doi:10.3390/microorganisms10030546_

Round 1

Reviewer 1 Report

This work fits the scope of the journal and represents fine efforts to describe the state of the art knowledge on picocyanobacteria

Section 2.2 is too short and does not include sufficient information. I suggest extending the description of cell surface functional groups and biomineralization capacity. Note that in addition to CaCO3 precipitation (i.e., Dittrich, M., Müller, B., Mavrocordatos, D., Wehrli, B., 2003. Induced calcite precipitation by cyanobacterium Synechococcus. Acta Hydroch. Hydrob. 31, 162–169.  Dittrich, M., Obst, M., 2004. Are picoplankton responsible for calcite precipitation in lakes? Ambio 33, 559–564.), there were studies of hydrous Mg carbonate precipitation in the presence of these bacteria (Mavromatis, V., Pearce, C.R., Shirokova, L.S., Bundeleva, I.A., Pokrovsky, O.S., Bénézeth, P., Oelkers, E.H., 2012. Magnesium isotope fractionation during hydrous magnesium carbonate precipitation with and without cyanobacteria. Geochim. Cosmochim. Acta 76, 161-174; Shirokova, L.S., Mavromatis, V., Bundelava, I.A., Pokrovsky, O.S., Bénézeth, P., Gérard, E., Pearce, C.R., Oelkers, E.H., 2013. Using Mg isotopes to trace cyanobacterially mediated magnesium carbonate precipitation in alkaline lakes. Aquat. Geochem. 19, 1-24); mineral dissolution kinetics (DOI: 10.1016/j.mineng.2014.01.019). May be it is worth organizing a separate section in the manuscript on biomineralization.

Other specific comments:

L188-192 check wording

L224 the Baikal Lake

Fig 5: uncertainties are missing

Fig 6: Insert is poorly visible; uncertainties son cell number are needed

Section 3.2 Please present the proof that cells are metabolizing at these conditions, and not simply dormant. Is it possible to measure their productivity, respiration or photosynthetic activity?

L443-455 : Present results of laboratory studies of Synechoccous culturing under these hostile conditions. Otherwise how do we know that these occurrences are not simple results of sea hydrological pattern and in-depth occasional submersion of surface waters (i.e., that these caynobacteria were physically moved to the deep horizons with water masses and simply survived there in a dormant state with metabolism as they would do in other hostile conditions)

L461-462 Provide a reference

Author Response

Rev 1

This work fits the scope of the journal and represents fine efforts to describe the state of the art knowledge on picocyanobacteria

Section 2.2 is too short and does not include sufficient information. I suggest extending the description of cell surface functional groups and biomineralization capacity. Note that in addition to CaCO3 precipitation (i.e., Dittrich, M., Müller, B., Mavrocordatos, D., Wehrli, B., 2003. Induced calcite precipitation by cyanobacterium Synechococcus. Acta Hydroch. Hydrob. 31, 162–169.  Dittrich, M., Obst, M., 2004. Are picoplankton responsible for calcite precipitation in lakes? Ambio 33, 559–564.), there were studies of hydrous Mg carbonate precipitation in the presence of these bacteria (Mavromatis, V., Pearce, C.R., Shirokova, L.S., Bundeleva, I.A., Pokrovsky, O.S., Bénézeth, P., Oelkers, E.H., 2012. Magnesium isotope fractionation during hydrous magnesium carbonate precipitation with and without cyanobacteria. Geochim. Cosmochim. Acta 76, 161-174; Shirokova, L.S., Mavromatis, V., Bundelava, I.A., Pokrovsky, O.S., Bénézeth, P., Gérard, E., Pearce, C.R., Oelkers, E.H., 2013. Using Mg isotopes to trace cyanobacterially mediated magnesium carbonate precipitation in alkaline lakes. Aquat. Geochem. 19, 1-24); mineral dissolution kinetics (DOI: 10.1016/j.mineng.2014.01.019). May be it is worth organizing a separate section in the manuscript on biomineralization.

REPLY: We agree with rev 1’s comment. While we cannot for space reason extend this section, we substituted the reference Dittrich et al 2005 that I cited in the test with the more appropriate Dittrich et al 2003, as suggested by the rev1 (unfortunately leaving out the other articles by Dittrich due to word count limitations).

We also added a sentence on the hydrous Mg carbonate precipitation citing the paper by Shirokova et al 2013 that refers to Synechococcus, but I did not add the paper by Mavromatis et al 2012 (that refers to Gloeocapsa).

Other specific comments:

L188-192 check wording

REPLY: we checked the paragraphs

L224 the Baikal Lake

REPLY: looking at the papers published on Lake Baikal the name in the manuscript is reported correctly

Fig 5: uncertainties are missing

REPLY: this figure is modified from another already published on ISME Journal. The counts were performed using flow cytometry and the the specifications of the method used are found in the paper Callieri et al 2019 ISME J. In any case we changed the figure 5 and 6 and added another figure.

Fig 6: Insert is poorly visible; uncertainties on cell number are needed

REPLY: we added a single figure (6) with more information

Section 3.2 Please present the proof that cells are metabolizing at these conditions, and not simply dormant. Is it possible to measure their productivity, respiration or photosynthetic activity?

REPLY: We did most of the measurements with strains isolated alive from the 750m layer (as explained in the paper on ISME J Callieri et al 2019). As this is a review, we did not offer new data, but only what has already been accepted by the scientific community and published.

L443-455 : Present results of laboratory studies of Synechoccous culturing under these hostile conditions. Otherwise how do we know that these occurrences are not simple results of sea hydrological pattern and in-depth occasional submersion of surface waters (i.e., that these caynobacteria were physically moved to the deep horizons with water masses and simply survived there in a dormant state with metabolism as they would do in other hostile conditions)

REPLY: In this review we simply reported what already published (Callieri et al 2019, Cabello Yeves et al 2021, Di Cesare et al 2020). Furthermore, we did nanosims experiments and trascriptomics (soon to be published) that demonstrate that the strains isolated in the Black Sea fermentate in dark anoxic conditions.

L461-462 Provide a reference

REPLY: we added the reference

Reviewer 2 Report

The authors provide a short review of the current understanding of the smallest prokaryotic phytoplankton, from their physiological features to their role and dynamics in different aquatic environments. Special attention is paid in the manuscript to the poorly understood ability of picocyanobacteria to adapt to highly adverse conditions, in particular, the unexpected recovery of Synechococcus in the deep dark anoxic layers of the Black Sea. The authors illustrate how adaptation to narrow ecological niches interacts with gene evolution and metabolic capacity of the microorganisms and, additionally, provide better explanations of the so-called “deep red fluorescence” signal which can be associated with the picocyanobacteria populations in deep dark anoxic waters. This idea is just the basis of the MS, shaping its structure and the major authors’ conclusions.

  • As the legend of Figure 3 mentions PBS, the image has to contain the designation of PBS. The legend does not also explain other abbreviations (AP, Chl, PSI, PSII, Cyt b6f).
  • Are there any reasons to provide an illustration of the picophytoplankton vertical distribution in two figures, Figs. 5 and 6, instead of combining them?
  • Some explanations should be provided to the cytogram in Fig. 6: axis and cluster descriptions, which station is represented.
  • Are 100 to 300 cells per ml enough to reliably measure the abundance using flow cytometry? It looks too unreliable. But this question is rather for the original paper.

Author Response

Rev 2

The authors provide a short review of the current understanding of the smallest prokaryotic phytoplankton, from their physiological features to their role and dynamics in different aquatic environments. Special attention is paid in the manuscript to the poorly understood ability of picocyanobacteria to adapt to highly adverse conditions, in particular, the unexpected recovery of Synechococcus in the deep dark anoxic layers of the Black Sea. The authors illustrate how adaptation to narrow ecological niches interacts with gene evolution and metabolic capacity of the microorganisms and, additionally, provide better explanations of the so-called “deep red fluorescence” signal which can be associated with the picocyanobacteria populations in deep dark anoxic waters. This idea is just the basis of the MS, shaping its structure and the major authors’ conclusions.

    As the legend of Figure 3 mentions PBS, the image has to contain the designation of PBS. The legend does not also explain other abbreviations (AP, Chl, PSI, PSII, Cyt b6f).

REPLY: we added the explanation in the legend and the indication of PBS.

    Are there any reasons to provide an illustration of the picophytoplankton vertical distribution in two figures, Figs. 5 and 6, instead of combining them?

REPLY: we agree with the referee and combined the figures forming a new fig. 5. Then we added Fig. 6 with 4 cytograms and Fig. 7 with the pictures taken at the epifluorescence microscopy.

    Some explanations should be provided to the cytogram in Fig. 6: axis and cluster descriptions, which station is represented.

REPLY: we added one specific figure (6).

    Are 100 to 300 cells per ml enough to reliably measure the abundance using flow cytometry? It looks too unreliable. But this question is rather for the original paper.

REPLY: Actually the number are low in the east basin, but when we analysed the metagenome, Synechococcus was present and we were able to count and to visualize picocyanobacteria with phycoerythrin in the deep part of the Black sea.

Reviewer 3 Report

microorganisms-1605459

The “dark side” of picocyanobacteria: life as we don’t know it (yet)

General comments:

The present study provides a review about the current knowledge of picocyanobacteria, focusing on the genus Synechococcus, in terms of its phylogenetics, physiological features and role and distribution. Furthermore, the authors present a case study related to the occurrence of Synechococcus in the mesopelagic environment of the Black Sea, revealing their ability to perform and survive in dark anoxic conditions.

Overall, I really appreciated reading the manuscript as it is well written and structured, therefore I recommend its publication. However, I still have a few suggestions to improve the manuscript, that are described below in the specific comments.

Specific comments:

Abstract

Page 1, Line 17: I suggest changing to: “Here, we briefly…”

Page 1, Line 18: I suggest changing to: “… focussing particularly…”

  1. The genus Synechococcus

2.1. Phylogenetics: a cryptic genus?

Page 2, Line 97: I would write the full name of the species as it is the first time you mention it: Synechococcus lacustris.

Page 3, Lines 100-106: I believe this sentence is too long, maybe split in two.

Page 3, Line 107: I would write the full name of the species as it is the first time you mention it: Synechococcus elongatus.

2.2. Physiological features

Page 6, Line 161: You can use the abbreviations PE and PC in this sentence, as you already mention the meaning in the previous sentence.

Page 6, Line 180: I suggest changing to: “…where green and blue-green light are available…”

2.3. Role and distribution

Page 7, Line 206: I suggest changing to: “…have focused on the temporal dimension…”

2.3.1 Temperature

Page 8, Line 231: I suggest changing to: “…below 0 ºC…”

Page 8, Line 238: I suggest changing to: “…cold bathypelagic Black Sea (around 1000 m depth)…”

2.3.3 Primary production

Page 10, Line 308: Correct to: “…Lake Maggiore and Lake Constance, respectively…”

  1. In the dark: Synechococcus surprising adaptability

3.1. A case study: the mesopelagic zone of the Black Sea

Page 12, Line 386: Define CDOM.

3.2. Unexpected metabolisms

Page 13, Lines 432-439: I would also split this sentence in two, as it is too long.

Author Response

Rev 3

General comments:

The present study provides a review about the current knowledge of picocyanobacteria, focusing on the genus Synechococcus, in terms of its phylogenetics, physiological features and role and distribution. Furthermore, the authors present a case study related to the occurrence of Synechococcus in the mesopelagic environment of the Black Sea, revealing their ability to perform and survive in dark anoxic conditions.

Overall, I really appreciated reading the manuscript as it is well written and structured, therefore I recommend its publication. However, I still have a few suggestions to improve the manuscript, that are described below in the specific comments.

Specific comments:

Abstract

Page 1, Line 17: I suggest changing to: “Here, we briefly…”

REPLY done

Page 1, Line 18: I suggest changing to: “… focussing particularly…”

REPLY done

    The genus Synechococcus

2.1. Phylogenetics: a cryptic genus?

Page 2, Line 97: I would write the full name of the species as it is the first time you mention it: Synechococcus lacustris.

REPLY done

Page 3, Lines 100-106: I believe this sentence is too long, maybe split in two.

REPLY done

Page 3, Line 107: I would write the full name of the species as it is the first time you mention it: Synechococcus elongatus.

REPLY done

2.2. Physiological features

Page 6, Line 161: You can use the abbreviations PE and PC in this sentence, as you already mention the meaning in the previous sentence.

REPLY done

Page 6, Line 180: I suggest changing to: “…where green and blue-green light are available…”

REPLY done

2.3. Role and distribution

Page 7, Line 206: I suggest changing to: “…have focused on the temporal dimension…”

REPLY done

2.3.1 Temperature

Page 8, Line 231: I suggest changing to: “…below 0 ºC…”

REPLY done

Page 8, Line 238: I suggest changing to: “…cold bathypelagic Black Sea (around 1000 m depth)…”

REPLY we changed to (around 1000m deep) for language reasons

2.3.3 Primary production

Page 10, Line 308: Correct to: “…Lake Maggiore and Lake Constance, respectively…”

REPLY done

    In the dark: Synechococcus surprising adaptability

3.1. A case study: the mesopelagic zone of the Black Sea

Page 12, Line 386: Define CDOM.

REPLY done

3.2. Unexpected metabolisms

Page 13, Lines 432-439: I would also split this sentence in two, as it is too long.

REPLY done

Round 2

Reviewer 1 Report

The authors adequately revised the manuscript